# Midwifery continuity of care: A scoping review of where, how, by whom and for whom?

**Billie F. Bradford**[1,2], **Alyce N. Wilson**[1,3], **Anayda Portela**[4], **Fran McConville**[4], **Cristina Fernandez Turienzo**[5], **Caroline S. E. Homer**[1] *

1 Maternal, Child, and Adolescent Health Program, Burnet Institute, Melbourne, Victoria, Australia, 2 Mater Research, University of Queensland, Brisbane, Queensland, Australia, 3 Melbourne Medical School, University of Melbourne, Melbourne, Victoria, Australia, 4 Department of Maternal, Newborn, Child, and Adolescent Health, World Health Organisation, Geneva, Switzerland, 5 Department of Women and Children's Health, Kings College London, London, United Kingdom

* Caroline.Homer@burnet.edu.au

**Data Availability Statement:** All data are in the Supporting information files.

**Funding:** This review was commissioned by the World Health Organization Department of Maternal,

## Abstract

Systems of care that provide midwifery care and services through a continuity of care model have positive health outcomes for women and newborns. We conducted a scoping review to understand the global implementation of these models, asking the questions: where, how, by whom and for whom are midwifery continuity of care models implemented? Using a scoping review framework, we searched electronic and grey literature databases for reports in any language between January 2012 and January 2022, which described current and recent trials, implementation or scaling-up of midwifery continuity of care studies or initiatives in high-, middle- and low-income countries. After screening, 175 reports were included, the majority (157, 90%) from high-income countries (HICs) and fewer (18, 10%) from low- to middle-income countries (LMICs). There were 163 unique studies including eight (4.9%) randomised or quasi-randomised trials, 58 (38.5%) qualitative, 53 (32.7%) quantitative (cohort, cross sectional, descriptive, observational), 31 (19.0%) survey studies, and three (1.9%) health economics analyses. There were 10 practice-based accounts that did not include research. Midwives led almost all continuity of care models. In HICs, the most dominant model was where small groups of midwives provided care for designated women, across the antenatal, childbirth and postnatal care continuum. This was mostly known as caseload midwifery or midwifery group practice. There was more diversity of models in low- to middle-income countries. Of the 175 initiatives described, 31 (18%) were implemented for women, newborns and families from priority or vulnerable communities. With the exception of New Zealand, no countries have managed to scale-up continuity of midwifery care at a national level. Further implementation studies are needed to support countries planning to transition to midwifery continuity of care models in all countries to determine optimal model types and strategies to achieve sustainable scale-up at a national level.

Newborn, Child and Adolescent Health and Ageing and funded through a grant received from Merck Sharp and Dohme Corp (MSD). CFT is supported by the National Institute for Health Research (NIHR) Applied Research Collaboration (ARC) South London, a NIHR Global Health Research Group (NIHR133232) and a NIHR Development and Skills Award (NIHR301603). CSEH is supported by an Australian National Health and Medical Research Council Fellowship (APP1137745). The funders had no role in study design, data collection and analysis, decision to publish, or preparation of the manuscript.

**Competing interests:** The authors have declared that no competing interests exist.

## Introduction

Continuity of care is a concept rooted in primary care involving the care of individuals (rather than populations) over time by the same care provider. It encompasses relational continuity, informational continuity and management continuity [1]. In the primary care setting, continuity of care has been shown to reduce mortality and hospitalisations, and increase patient satisfaction [2]. Continuity of care also has an important place in chronic care settings, such as palliative care [3].

In the maternal and newborn care setting, midwife-led continuity of care refers to a model whereby care is provided by the same midwife, or small team of midwives, during pregnancy, labour and birth, and the postnatal periods with referral to specialist care as needed [4]. Midwife-led also refers to a model of care which is provided there is a distinct occupational group of midwives [5] and the person is fully qualified, regulated and deployed only as a midwife. This contrasts to systems in many countries (most countries in Africa for and South East Asia for example) where nurse-midwives are rotated to either nursing or midwifery duties. Midwife-led continuity models in a small number of HICs have been associated with lower rates of preterm birth (24% reduction), and lower fetal loss before and after 24 weeks and neonatal deaths (16%) less likely to lose their babies overall (combined reduction in fetal loss and neonatal death) for women at low and mixed risk of complications compared to other models of care. In addition, women are less likely to experience interventions and more likely to report positive experiences of care [4]. A Cochrane review of reviews of interventions during pregnancy to prevent preterm birth also found that these models had clear benefit in reducing preterm birth and perinatal death [6]. Women prefer the personalised experience provided by such models, leading to trust between midwife and woman and empowerment of both women and midwives [7].

Models of care that provide continuity across the childbearing continuum are complex interventions, and the pathway of influence that produces these positive outcomes is unclear. A number of plausible hypotheses require further investigation. For example, it could be that midwives provide a mechanism that enables effective and equitable care to be provided by better coordination, navigation and referral; and/or that relational continuity and advocacy engenders trust and confidence between women and midwives, resulting in women feeling safer, less stressed and more respected [4]. Access to organisational infrastructure, innovative partnerships, and robust community networks has been found crucial to overcome barriers, address women's, newborns' and parents' needs and ensure quality of care [8].

Inequity is a key driver of adverse perinatal outcome, both between and within countries. Some observational studies of midwife-led continuity of care models in socially and economically disadvantaged populations in high-income countries (HIC) have reported significant reductions in pre-term birth and caesarean sections in diverse cohorts of women in the United Kingdom [9–12]. In Australia, a study of maternity care during significant floods in Queensland showed that midwife-led continuity of care mitigated the social and emotional impacts of the floods [13]. Another Australian study showed reduced preterm births amongst Australian Aboriginal and Torres Strait Islander women who received midwife-led continuity of care [14]. These studies suggest that women who typically experience a greater burden of adverse perinatal outcome, may derive greater benefit from continuity of care. However, understanding how continuity per se may mitigate inequities in maternal and newborn health remains a research priority.

Despite evidence supporting midwife-led continuity of care and guidelines from the World Health Organization which recommend midwife-led continuity-of-care models for pregnant women in settings with well-functioning midwifery programmes [15–17] only a small

proportion of women internationally have access to such care. The current evidence suggests that access to midwife-led continuity of care models is largely confined to a small number of HICs notably Australia, Canada, New Zealand and the United Kingdom [17] where a distinct occupational group of midwives has been a central part of the health systems for decades. Barriers to implementation of midwifery-led continuity of care exist across all country income levels and include a lack of local health system financing, shortage of personnel including administrative and other support staff [4]. It is not clear to what extent midwife-led continuity of care has been implemented in low- to middle-income countries (LMIC). Many LMICs have a model of predominantly nurse-midwives who are deployed to both nursing and midwifery duties, often preventing midwife-led continuity of care models. Advancing understanding around which countries have implemented continuity of care models for maternal and newborn health, how, for whom, and in what context, is crucial for successful implementation, scale-up and sustainability.

The overall aim of this review was to understand the global implementation of midwifery continuity of care, asking the questions: Where, how, by whom and for whom are midwifery continuity of care initiatives implemented?

## Materials and methods

A scoping review was undertaken guided by the approach described by Arksey and O'Malley [18] and further defined by Levac and colleagues [19]. The following five steps were followed: i) identifying the research question; ii) identifying the relevant literature; iii) study selection, iv) charting the data; and v) collating, summarising and reporting the results.

We used the broad definition of midwifery from The Lancet Series on Midwifery as our starting point, that is, "skilled, knowledgeable, and compassionate care for childbearing women, newborn infants and families across the continuum from pre-pregnancy, pregnancy, birth, postpartum and the early weeks of life [20]. Midwifery continuity of care was defined as care delivered by the same known care provider or care provider team across two or more parts in the care continuum–antenatal, intrapartum, postnatal and neonatal periods. In some settings, continuity of care may be provided by cadre other than midwives, for example, nurses or physicians. Thus, eligible papers could include care providers that were midwives and non-midwives, such as, nurses, community health workers and physicians. We excluded reports on care primarily by traditional birth attendants (TBA).

### Identifying the relevant literature—Search strategy and selection criteria

In order to develop the search strategy, a preliminary search of PubMed and Google scholar using the terms 'midwifery or midwife-led continuity of care' were used to locate key systematic and scoping reviews on the topic and identify relevant search terms for the systematic search strategy (see S1 Text for the search strategy). We then searched the following electronic databases: MEDLINE, CENTRAL, CINAHL, PsychINFO and Web of Science. A subject librarian reviewed search terms, keywords and strategies. In addition, we searched PubMed, Google Scholar, PROSPERO, Scopus and Dimensions and the WHO International Clinical Trials Registry platform. We conducted the search on the 20 February 2022 and included publications (peer reviewed studies and reports) in the past 10 years.

A key area of interest was implementation of continuity of midwifery care in LMICs but we recognised that reports of implementation may be published in formats other than peer reviewed publications. Eligible papers therefore included implementation studies or reports of implementation of midwifery continuity of care in the grey literature. We sourced grey literature through online searches on the websites of relevant professional groups, United Nations

agencies and non-government organisations (NGO). We circulated a call for relevant materials through online list servs (email groups) and through midwifery contacts. The International Confederation of Midwives assisted by emailing all member associations asking for any relevant reports.

Eligible reports could report on midwifery continuity of care efforts in HICs and LMICs. Reports from implementation efforts by government programmes, private providers, professional organisations, NGOs and universities and research studies of any design were eligible for inclusion. Protocols that reported studies that were underway, but not concluded, were also eligible. Opinion pieces, editorials and other materials, which included details of midwifery continuity of care initiatives, were also eligible. Publications in any language were eligible. The search was limited to reports published in the last ten years (January 2012 to January 2022) to ensure the information was contemporary and therefore of greatest relevance to policy makers.

Reports identified through both peer-reviewed and grey literature databases were hand-searched for other potentially relevant studies. These included reference lists of relevant systematic reviews, and published conference abstracts, as well as any reports forwarded to authors in response to a call for notification of new or ongoing initiatives from key global stakeholder organisations, such as the International Confederation of Midwives.

Reports were excluded they if reported on midwifery continuity of care in general but did not report on a continuity of care practice initiative. We excluded systematic and literature reviews although their reference lists were searched for relevant primary studies.

## Study selection

All reports identified through database searching were imported into Endnote referencing programme (Endnote 20, Clarivate Analytics, Philadelphia), and duplicates removed. Remaining citations (n = 5789) were uploaded into systematic review software Covidence (Covidence 2022, Veritas Health Innovations, Melbourne). Two authors independently conducted initial title and abstract screening and undertook full-text review. A third author screened a random selection of 10% of studies and discrepancies were discussed and resolved.

## Charting the data

The following information was extracted for all included reports: country, income-level (as defined by the World Bank [21], study design (if applicable), setting (urban/rural and community-based or facility-based), novel or scaled-up initiative, model of care, level of continuity (antenatal and intrapartum, antenatal and postnatal, intrapartum and postnatal) and cadre of care providers, (e.g. mix of providers involved). We also collected information on the inclusion of priority population groups–these are groups of people who are persistently disadvantaged by existing systems of power with demographic features known to be associated with adverse perinatal outcomes, such as ethnic minorities, urban and remote women, socially disadvantaged, and Indigenous women. We have described these specific groups as priority rather than vulnerable populations [22]. Reporting of the scoping review findings follows the PRISMA-ScR (Preferred Reporting Items for Systematic reviews and Meta-Analyses extension for Scoping Reviews) format (see S1 Checklist) and reference (Fig 1) [23]. Appraisal of study quality or meta-analysis was not undertaken.

## Results

In total, 6595 references were identified from electronic peer-reviewed databases, 821 duplicate records were removed prior to uploading to Covidence, a further 634 duplicates were removed

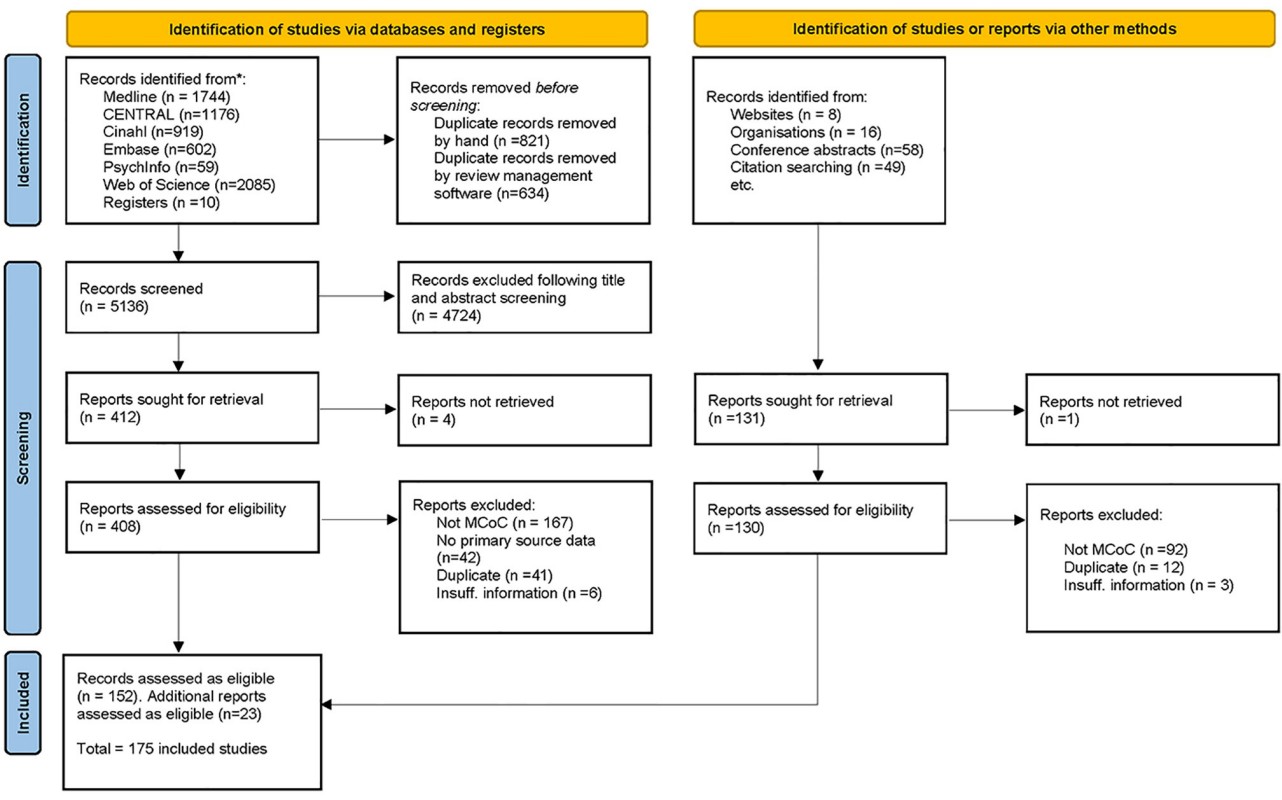

**Fig 1. PRISMA diagram.**

by automation. Of the 5136 remaining references, 4728 did not meet the inclusion criteria. A further 256 references were excluded at the full-text stage as either: they did not describe continuity of care according to our pre-determined criteria (167); did not include primary source data (42); were duplicates (41); or did not have insufficient detail regarding the model of care (6). One hundred and fifty-two (152) peer-reviewed publications were eligible based on inclusion and exclusion criteria. A further 23 reports were identified following the grey literature search, bringing the total to 175 (Fig 1). Details are listed in S1 Table.

Of the 175 individual reports, 152 (86.8%) were peer reviewed publications, 18 (10.3%) were conference abstracts, and the remaining five (3%) were published or unpublished reports. Reports primarily reported on birth outcomes (n = 54, 31%), women's (including some partners') views and experiences (n = 47, 27%) and midwives (including doctors) views and experiences (n = 33, 19%). There were 18 reporting on the model of care more broadly, including implementation challenges (n = 18, 10%), and 14 (7%) that were focussed on the experience of midwifery students providing continuity of care as part of their education. The majority of these student-focused reports were from Australia. Fewer reports focussed on the experience of midwifery managers (n = 3, 2%), while four were cost analyses (2%).

There were 163 unique studies including eight (4.9%) randomised or quasi randomised trials, 58 (38.5%) qualitative, 53 (32.7%) quantitative (cohort, cross sectional, descriptive, observational), 31 (19.0%) survey studies, and three (1.9%) health economics analyses. There were 10 practice-based accounts that did not include research.

## 'The where': Country and setting

Of the 175 included reports, the majority (n = 157, 90%) were from HICs and 18 (10%) from LMICs (Table 1). Most were from Australia, (n = 71, 41%), followed by the United Kingdom (England, Scotland, Wales) (24, 14%), Sweden (13, 8%), Canada (8, 5%), Denmark (6, 4%), New Zealand (7, 4%), Japan (5, 3%), with less than five reports described initiatives conducted in Belgium, Finland, Germany, Greece, Ireland, Netherlands, Norway, Singapore, Switzerland and the United States of America (USA). In the LMICs, three from Palestine, three from China, two each from Bangladesh and Indonesia, and one from each of the remaining countries.

Overall, most midwifery continuity of care models were based in urban areas (n = 126, 72%). In HIC, three-quarters of services (n = 118, 75%) were urban-based, whereas in LMICs just under half (n = 8, 44%) were urban-based. Hospital or facility-based services were most common across all income levels (n = 124, 72% overall).

## 'The how': Describing the way continuity of care is provided

There were a number of different terms used to define the model of care, and the level of continuity provided across the continuum of care varied with no single term used. Overall, the most common terms were caseload midwifery (n = 63, 36%), midwifery-led continuity (n = 60, 34%), or team/midwifery group practice (n = 40, 23%). Most described services designed so that the same providers provided care across the continuum–antenatal, intrapartum and postnatal (n = 159, 91%). There were eight which described continuity only across the antenatal and postpartum periods [24–31] (excluding labour and birth), and five reported [32–36] (including 3 unique examples) where the continuity was provided only across antenatal and intrapartum periods without postpartum care.

In HICs, the most dominant approach is where small groups of midwives provide care for designated women, known as caseload midwifery or midwifery group practice in countries like Australia [37], the United Kingdom [9], Denmark [38], Sweden [39], and Singapore [40] where the number of midwives is usually two to four. In other countries, for example, Japan [41] and Switzerland [42], the approach is also called team midwifery and the number of midwives is five or more.

The continuity of care services were located as part of the usual hospital [37, 43], in an alongside birth centre [44–46] or in a free-standing birth centre [47]. Some midwife-led continuity of care services were offered through homebirth practices, either as part of the hospital system [48, 49] or as a private service [50]. Most services were based in urban areas but there were some examples from rural areas in Australia [51–53], Sweden [54–56] and Scotland [57] (Table 2).

Although midwife-led continuity of care was available in a number of countries, mostly high-income with a cadre of midwives, in select facilities and locations, it was generally not scaled-up nationally. The exception being New Zealand, where the Lead Maternity Carer model is national allowing to midwives provide continuity of care to all women regardless of risk, in either caseloading or small community-based group practices, under a national funding arrangement and with medical or other collaboration when required [58].

In LMICs there was greater diversity in structure of arrangements for provision of midwifery continuity of care models. Models of care included a lead midwife delivering care across the continuum [59, 60], midwives on-call for women during labour who they had previously seen for antenatal care [61], and a midwifery continuity of care team that ran in parallel with an obstetric team [62]. An initiative in Ethiopia involved the same midwife providing antenatal, intrapartum and postnatal care to the same women [63, 64]. An initiative in Kenya

**Table 1. Midwifery continuity of care publications by country.**

| Country | Income level as defined by World Bank [21] | Publications by country (n) | % |
|---|---|---|---|
| Australia | High | 71 | 40.6% |
| United Kingdom | High | 24 | 13.7% |
| Sweden | High | 13 | 7.4% |
| Canada | High | 8 | 4.6% |
| New Zealand | High | 7 | 4.0% |
| Denmark | High | 6 | 3.4% |
| Japan | High | 5 | 2.9% |
| Norway | High | 4 | 2.3% |
| China | Upper middle | 3 | 1.7% |
| Netherlands | High | 3 | 1.7% |
| Palestine | N/A | 3 | 1.7% |
| Switzerland | High | 3 | 1.7% |
| USA | High | 3 | 1.7% |
| Bangladesh | Lower middle | 2 | 1.1% |
| Ethiopia | Low | 2 | 1.1% |
| Indonesia | Lower middle | 2 | 1.1% |
| Ireland | High | 2 | 1.1% |
| Pakistan | Lower middle | 2 | 1.1% |
| Singapore | High | 2 | 1.1% |
| Afghanistan | Low | 1 | 0.6% |
| Belgium | High | 1 | 0.6% |
| Finland | High | 1 | 0.6% |
| Germany | High | 1 | 0.6% |
| Ghana | Lower middle | 1 | 1.7% |
| Greece | High | 1 | 0.6% |
| Iran | Lower middle | 1 | 0.6% |
| Kenya | Lower middle | 1 | 0.6% |

*Due to rounding, percentages may be >100%

involved midwifery care across the childbearing continuum, embedded within a family planning and HIV care service [65]. One initiative in China [66] facilitated continuity of care for women wishing to have a vaginal birth, where efforts were made for women to see the same midwife for intrapartum and postnatal care. In the Palestinian initiative [67–69], midwives were allocated geographical areas to provide antenatal and postnatal care for between 50–100 women. One initiative in Bangladesh [70] and another in Iran [71] involved teams of midwives providing care in a private midwifery clinic associated with two local hospitals.

Similar to HICs, continuity of care services were located as part of the usual hospital services (eg in Pakistan [59, 72], China [61], Ethiopia [63, 64], Palestine [67–69] or in community health centres or maternity clinics, for example in Bangladesh [60, 70], Kenya [65] and Afghanistan [73]. Most services were based in urban or semi-urban areas but there were some examples from rural areas, for example, Palestine [67–69], Bangladesh [70], Afghanistan [73] and Indonesia [74].

Reports from China [62], Ethiopia [63], Iran [71], Kenya [65] and Pakistan [59] provided some degree of continuity of care across all antenatal, intrapartum and postnatal periods. Two reports, one from China [61] and the one from Kenya [65] provided care across antenatal and intrapartum. A study in China [66] involved the provision of midwife-led care at antenatal,

**Table 2. Midwifery continuity of care: Where, how, by whom and for whom by income level.**

| | All reports N = 175 (%) | | High-income countries N = 157 (%) | | Low- or middle-income countries N = 18 (%) | |
|---|---|---|---|---|---|---|
| **Type of publication** | n = 175 | | n = 157 | | n = 18 | |
| Peer reviewed paper | 152 | 86.8% | 135 | 86.0% | 17 | 94.4% |
| Conference Abstract | 18 | 10.3% | 18 | 11.5% | | |
| Published report | 4 | 2.3% | 3 | 1.9% | 1 | 5.5% |
| Unpublished report | 1 | 0.6% | 1 | 0.6% | | |
| **Design or approach–unique studies only (N = 162)** | | | (n = 145) | | (N = 18) | |
| Qualitative study | 58 | 35.6% | 52 | 35.9% | 6 | 33.3% |
| Quantitative (cohort, cross sectional, descriptive, observational) | 53 | 32.5% | 47 | 32.4% | 6 | 33.3% |
| Survey study | 30 | 19.0% | 28 | 19.3% | 2 | 16.7% |
| Trial (randomised or quasi randomised) | 8 | 4.9% | 6 | 4.1% | 2 | 11.1% |
| Health economics analysis | 3 | 1.8% | 3 | 2.1% | | |
| Practice story (non-research) | 10 | 6.1% | 9 | 6.2% | 1 | 5.6% |
| **Focus of the study/project** | | | | | | |
| Women's outcomes | 54 | 30.9% | 47 | 29.9% | 7 | 38.9% |
| Women's experiences | 48 | 28.0% | 44 | 28.0% | 4 | 27.8% |
| Midwives' experiences | 33 | 19.4% | 30 | 19.1% | 3 | 16.7% |
| Model of care implementation | 18 | 10.3% | 16 | 10.2% | 2 | 11.1% |
| Midwifery students | 14 | 7.4% | 13 | 8.3% | 1 | 5.6% |
| Cost analysis | 4 | 2.3% | 4 | 2.5% | | |
| Managers | 3 | 1.7% | 3 | 1.9% | | |
| **WHERE** | | | | | | |
| Setting | | | | | | |
| Urban | 126 | 72.0% | 118 | 75.2% | 8 | 44.4% |
| Rural | 24 | 13.7% | 14 | 8.9% | 10 | 55.6% |
| Remote | 4 | 2.3% | 4 | 2.5% | | |
| Urban, rural, remote | 17 | 9.7% | 17 | 10.8% | | |
| Unknown | 4 | 2.3% | 4 | 2.5% | | |
| Location | | | | | | |
| Hospital based | 125 | 71.8% | 112 | 71.3% | 13 | 76.5% |
| Community | 35 | 20.1% | 32 | 20.4% | 3 | 17.6% |
| Birth centre | 13 | 7.5% | 12 | 7.6% | 1 | 5.9% |
| Unknown | 1 | 0.6% | 1 | 0.6% | | |
| **HOW** | | | | | | |
| Model of care | | | | | | |
| Caseload midwifery | 63 | 36.0% | 62 | 39.5% | 1 | 5.6% |
| Midwife-led continuity | 60 | 34.3% | 46 | 29.3% | 14 | 77.8% |
| Midwifery group practice | 40 | 22.9% | 40 | 25.5% | | |
| Caseload midwifery—private | 3 | 1.7% | 3 | 1.9% | | |
| Team midwifery | 3 | 1.7% | 2 | 1.3% | 1 | 5.6% |
| Midwife-led clinic | 1 | 0.6% | | | 1 | 5.6% |
| Continuity of care with other cadres | 5 | 2.9% | 4 | 2.6% | 1 | 5.6% |
| Level of continuity of care | | | | | | |
| Antenatal, intrapartum, postpartum | 159 | 91.4% | 144 | 91.7% | 15 | 88.2% |
| Antenatal, intrapartum | 7 | 4.0% | 5 | 3.2% | 2 | 11.8% |
| Antenatal, postpartum | 8 | 4.6% | 8 | 5.1% | | |
| **BY WHOM** | | | | | | |

*(Continued)*

**Table 2.** (Continued)

| | All reports N = 175 (%) | | High-income countries N = 157 (%) | | Low- or middle-income countries N = 18 (%) | |
|---|---|---|---|---|---|---|
| Primary provider | | | | | | |
| Midwives | 143 | 82.3% | 129 | 82.2% | 14 | 77.8% |
| Midwifery students | 16 | 9.1% | 15 | 9.6% | 1 | 11.1% |
| Midwives, Indigenous Health Worker | 5 | 2.9% | 5 | 3.2% | | |
| Midwives or doctors | 1 | 0.6% | | | 1 | 5.6% |
| Midwives, obstetricians | 4 | 2.3% | 3 | 1.9% | 1 | 5.6% |
| Midwives, GPs | 1 | 0.6% | 1 | 0.6% | | |
| Midwives and child health nurses | 1 | 0.6% | 1 | 0.6% | | |
| Midwives, social worker | 1 | 0.6% | 1 | 0.6% | | |
| Obstetricians | 1 | 0.6% | 1 | 0.6% | | |
| Public health nurse | 1 | 0.6% | 1 | 0.6% | | |
| **FOR WHOM** | | | | | | |
| Priority populations | | | | | | |
| No | 132 | 74.5% | 157 | 74.5% | 14 | 78.8% |
| Yes | 44 | 25.5% | 40 | 25.5% | 4 | 22.2% |
| Indigenous women | 13 | 7.4% | 13 | 8.3% | | |
| Socially or economically disadvantaged or women (priority groups) | 9 | 5.1% | 8 | 5.7% | 1 | 5.5% |
| Young/adolescent women | 4 | 2.3% | 4 | 2.5% | | |
| Specific risks of preterm birth | 2 | 1.1% | 2 | 1.3% | | |
| Ethnic minority or African American women | 2 | 1.1% | 2 | 1.3% | | |
| Women with drug or alcohol dependence problems | 2 | 1.1% | 2 | 1.3% | | |
| Rural or remote | 11 | 7.0% | 8 | 5.1% | 3 | 11.1% |
| Other | 2 | 1.1% | 2 | 1.3% | | |

intrapartum and postnatal time points, but continuity of care with the same/a known provider was only guaranteed at intrapartum and postnatal time points. S1 Table provides more details on each of the initiatives and S2 Table gives additional detail on models of care from LMICs.

## The 'by whom': Providers of midwifery continuity of care

Midwives were the dominant provider of continuity of care across all settings. Services were mostly midwife-led with some reports including other cadre as well. Integration with existing services including systems for referral to obstetric services when needed was usual.

In HIC, almost all models of continuity of midwifery care involved care provided by midwives and/or midwifery students. A small number included midwives and other cadre. For example, programs with midwives and Aboriginal Health Workers (Indigenous health providers) [14, 75–78]; collaborations with general practitioners, obstetricians or a social worker [44, 79–81]. Just two examples did not include midwives; a model in Finland where continuity of care is provided by a nurse who takes care of the family from the pregnancy until the child reaches school age [26, 82], and an example in Ireland [83], where continuity of care was provided by a privately practising obstetrician.

All except two of the continuity of care initiatives in LMICs were midwife-led. The initiative from Ghana [84] was provided by midwives, nurses and doctors while the one in Kenya [65] was provided by community based midwives who may have nursing or midwifery qualifications and other health professional with obstetric skills who reside in the community.

There were 16 reports which described midwifery students providing continuity of care, most of these were from Australia, Norway and Indonesia [74, 85–101]. Midwifery students were placed with women, providing continuity of care to a defined number of women over their education program, as a way to engage them in this model of care [91, 94].

### The 'for whom': Priority groups for continuity of care initiatives

Of the 175 initiatives, 44 (25.5%) of these were implemented for women and newborns with risk of adverse outcomes (Table 3). These included women from Indigenous communities, refugee and migrant populations, young mothers, women living in rural and remote areas, women who experience socioeconomic disadvantage, women with a history of substance abuse, chronic illness, and ethnic minority groups. The majority were from Australia [23], United Kingdom [9] and Canada [3]. There were four examples from LMICs, these were designed primarily for rural and remote communities in Palestine [67–69], Bangladesh [70], Afghanistan [73] and Kenya [84].

## Discussion

This scoping reviewed aimed to map where, how, by whom, and for whom are midwifery continuity of care models are being implemented globally. The majority of models identified were in HICs, largely in Australia and the United Kingdom. Notably, all countries where five or more continuity of midwifery care initiatives were identified in the last 10 years are high-income and provide free public healthcare to their citizens and have a distinct cadre of midwives which makes this possible (Australia, Canada, Denmark, Japan, New Zealand, Sweden, and United Kingdom). Only 18 initiatives were identified in LMICs.

There is a growing body of literature demonstrating beneficial effects of midwifery continuity of care [4, 8]. Midwifery continuity of care is a complex, multi-faceted intervention and teasing out which elements impart benefit to recipients of care is difficult. We found that

**Table 3. Initiatives for women, newborns and families with risk of adverse outcomes by country.**

| Country | Number of initiatives | Priority or vulnerable populations (n =) * |
|---|---|---|
| Australia | 23 | Rural and remote (6), Indigenous (11), Young mothers (3), Social disadvantage (3), Impacted by natural disaster (1), substance abuse (1) |
| Bangladesh | 1 | Tea Garden Workers (1) |
| Canada | 3 | Rural and remote (1), Indigenous (2), social disadvantage (1) |
| Denmark | 1 | Chronic conditions (1) |
| United Kingdom | 9 | High social risk (2), high social deprivation (4), young mothers (1), high preterm birth risk (1), refugee and migrant (1) |
| United States | 2 | African American women (1), opioid use disorders (1) |
| Greece | 1 | Refugee and migrant (1) |
| Kenya | 1 | Rural (1) |
| Netherlands | 1 | Refugee and migrant (1) |
| New Zealand | 2 | High social deprivation (2), ethnic-minority (1) |
| Palestine | 1 | Rural (1) |
| Sweden | 1 | Rural (1) |
| Afghanistan | 1 | Rural (1) |

*as described by the authors

Note: Some country totals are more than the country count as some studies addressed more than one priority population

almost all papers included in this review, involved continuity of care initiatives led by mid-wives or midwifery students (with midwife supervision). This was despite casting the net wide to identify continuity of care initiatives provided by any health provider across two or parts of the maternal and newborn care continuum.

Reviews of continuity of care in maternal and newborn care have focused on midwife-led continuity of care compared with other models of care such as doctor-led and shared care models [4, 102]. However, a previous integrative review of midwife-led care in LMICs, found that just over half of studies included in the review included only midwives, with other cadres of health professionals including nurses, nurse-midwives, doctors, traditional birth attendants and family planning workers [103]. Whilst there is scope for other non-midwife health provid-ers to provide continuity of care, such as family physicians [104]and community health work-ers [105], which may particularly be of value in LMICs countries where there is a shortage of midwives [106], there are few studies or reports available about these continuity of care models and their benefits. Although other cadre are not precluded from providing continuity of care, this review has shown that in the global literature, continuity of care across the maternal and newborn continuum is reported to be almost exclusively provided by midwives and is a signifi-cant area of quality improvement and research interest for midwives.

An encouraging finding from this review was the significant proportion of initiatives in HICs which focussed on women and newborns with vulnerabilities related to social and eco-nomic determinants of health (23.2%). The evidence that such initiatives are feasible for a diverse range of priority groups across many countries could demonstrate recognition of the benefits of continuity of care in improving outcomes for those with greater social and eco-nomic barriers to good health outcomes. This has implications for future research in that pre-vious studies exploring childbirth outcomes from midwifery continuity of care frequently involve low-risk women [4], or women who had self-selected to be part of a midwife-led care project and thus are more likely to experience a positive outcome. This review has revealed that initiatives in a range of settings involve groups acknowledged to be at increased risk of adverse outcome. Larger scale and robust studies of midwifery continuity of care initiatives involving populations who experience social and economic disadvantage, and/or are at increased obstetric risk are both feasible and desirable.

## Implications for policy, practice, research

This review has revealed that most studies, or reports, on midwifery continuity of care describe models led by midwives within HICs. Despite the benefits of midwife-led continuity of care, none of these countries has managed to scale-up this approach to being the standard of care at a national level, other than New Zealand. This highlights the organisational challenges of wide-spread implementation and the importance of system-level reform to enable countries to tran-sition to this model of care and to scale-up. This reform means having adequate funding, support to enable midwives to be educated, and regulated, to work to their full scope of prac-tice including flexibility and autonomy, self-managed time, team space, telephone access, and being able to work safely in the community and having access to transport and referral services [8, 107].

Fewer than 10% of initiatives included in this review were from LMICs and only one was a clinical trial. The greatest burden of maternal and newborn deaths and stillbirths exists in LMICs. In HICs, midwife-led continuity of care has potential to reduce preventable maternal and newborn mortality and morbidity and stillbirths, however system-level reform and ensur-ing an enabling environment is still key [44]. The lack of midwifery continuity of care initia-tives in LMICs, highlights the need for greater investment to ensure well-functioning

midwifery systems can be developed with monitoring, evaluation and research to understand the effect of different models and associated benefits and/or challenges in different contexts. Operational research that identifies the barriers, facilitators and blockages to implementing models of midwifery continuity of care is needed, including in settings where there are shortages of midwives. In order to facilitate transition to, and scale-up of, midwifery continuity of care in LMICs, key considerations include strengthening midwifery education and regulation and ensuring the presence of an enabling environment [66, 73].

Future systematic and scoping review studies would be enhanced by clear reporting of midwifery continuity model type, implementation details (including on midwife competence, scope of practice, deployment) and degree of continuity achieved within published studies and reports. Establishment of a classification system for this purpose would also enhance implementation efforts. One example of a classification system in a country which has an identifiable cadre of midwives is the Maternity Care Classification System (MaCCS) which was developed to classify, record and report data about maternity models of care in Australia [108, 109]. The MaCCS includes a series attributes including the target groups, profession of provider, the caseload size, the extent of planned continuity of care and the location of care to come up with 11 major model categories (see S3 Table for details) [110]. This classification system is now being included in all routine data systems in Australia so that, in the future, outcomes by model of care will be reported. While this is developed for one high-income country, an adaptation for global utility could be useful.

Measuring the extent to which continuity of care is achieved is the second key area. The health insurance industry in the USA has developed measures to assess patterns of visits to providers and therefore, the level of continuity of care [111]. The measures include the Bice-Boxerman Continuity of Care Index (measures the degree of coordination required between different providers during an episode), the Herfindahl Index (the degree of coordination required between different providers during an episode), the Usual Provider of Care (the concentration of care with a primary provider) and Sequential Continuity of Care Index (the number of handoffs of information required between providers). The Usual Provider of Care index has also been used to assess continuity of care in general practice in the UK, that is, to assess the proportion of a patient's contacts that was with their most regularly seen doctor [112]. For example, if a patient had 10 general practitioner contacts, including six with the same doctor, then their usual provider of care index score would be 0.6. With the exception of one study, none of the papers in this review had applied such indexes. This is an important consideration for the future.

## Strengths and limitations

This review provides a summary of midwifery continuity of care efforts globally. As countries look to strengthen midwifery and quality of care for women and newborns during pregnancy, childbirth and postnatal periods, understanding implementation in all resource settings is important. In this review the broad criteria for inclusion allowed for identifying the maximum number of implementation efforts in LMICs to be identified. Despite the efforts to reach out, and although no language filters were applied, search terms were in English thus we may have missed some ongoing efforts. We also did not measure, or account for, the skills and competencies in the different cadres providing care, or if they are always deployed as midwives or provide details about the profile/qualifications of the healthcare providers, the way the midwifery system function, if any affiliation to healthcare centres, support systems, health costs and coverage or safety outcome indicators as these were reported differently or not at all across the papers. Finally, in this review we were not able to reliably determine the extent to which

women receiving care were able to see the same individual care provider. Relational continuity is a key element of continuity of care, and possible mechanism for beneficial effects, which requires repeat contact over time between individual care providers and recipients of care.

## Conclusions

This review mapped midwifery continuity of care initiatives globally. The majority of initiatives identified were in HICs, with fewer identified in LMICs. Almost all initiatives identified in LMICs were led by midwives (some of whom worked in a model in which they were also deployed as nurses), despite our efforts to identify models led by other skilled health professionals. Almost no countries have managed to scale-up midwifery continuity of care to being the standard of care at a national level. This highlights the organisational challenges of widespread implementation and the importance of system-level reform to enable these models of care to scale-up. Nevertheless, examples of successful implementation of midwifery continuity of care in low-resource settings reported show that advances in this area are possible.

A number of initiatives identified in HICs focused on women and newborns at risk of adverse outcomes, demonstrating the value of midwifery continuity of care in populations who experience social and economic disadvantage and vulnerabilities. There is a need for further research on midwifery continuity of care models in LMICs, and strategies to facilitate transition to, and scale-up of, midwifery continuity of care initiatives globally.

## Supporting information

**S1 Text. Search strategy.**
(DOCX)

**S1 Table. All included items.**
(DOCX)

**S2 Table. Additional details from low- and middle-income countries.**
(DOCX)

**S3 Table. Major model categories in MaCCS.**
(DOCX)

**S1 Checklist. Preferred reporting items for systematic reviews and meta-analyses extension for scoping reviews.**
(DOCX)

## Acknowledgments

Thank you to Rana Islamiah Zahroh, PhD student and researcher at the University of Melbourne in Australia for assistance mapping the data. Thanks also to Rosemary Rowe, Subject Librarian at Faculty of Health, Victoria University of Wellington in New Zealand and to Allisyn Moran and Joao Paolo Souza (WHO) for useful feedback and advice.

## Author Contributions

**Conceptualization:** Anayda Portela, Fran McConville, Caroline S. E. Homer.

**Data curation:** Billie F. Bradford, Alyce N. Wilson, Caroline S. E. Homer.

**Formal analysis:** Billie F. Bradford, Alyce N. Wilson, Cristina Fernandez Turienzo, Caroline S. E. Homer.

**Funding acquisition:** Anayda Portela, Fran McConville, Caroline S. E. Homer.

**Investigation:** Caroline S. E. Homer.

**Methodology:** Billie F. Bradford, Anayda Portela, Fran McConville, Cristina Fernandez Turienzo, Caroline S. E. Homer.

**Project administration:** Billie F. Bradford, Caroline S. E. Homer.

**Supervision:** Anayda Portela, Caroline S. E. Homer.

**Validation:** Billie F. Bradford, Alyce N. Wilson, Anayda Portela, Cristina Fernandez Turienzo, Caroline S. E. Homer.

**Writing – original draft:** Billie F. Bradford.

**Writing – review & editing:** Billie F. Bradford, Alyce N. Wilson, Anayda Portela, Fran McConville, Cristina Fernandez Turienzo, Caroline S. E. Homer.

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
