## [Decision Letter · Decision Letter 0]

19 Jul 2022

PGPH-D-22-00905

Midwifery continuity of care: A scoping review of where, how, by whom and for whom?

Dear Dr. Homer,

Thank you for submitting your manuscript to PLOS Global Public Health. After careful consideration, we feel that it has merit but does not fully meet PLOS Global Public Health’s publication criteria as it currently stands. Therefore, we invite you to submit a revised version of the manuscript that addresses the points raised during the review process.

We look forward to receiving your revised manuscript.

Kind regards,

Ahmed Waqas

Academic Editor

Journal Requirements:

a. Please clarify all sources of funding (financial or material support) for your study. List the grants (with grant number) or organizations (with url) that supported your study, including funding received from your institution. 

b. State the initials, alongside each funding source, of each author to receive each grant.

c. State what role the funders took in the study. If the funders had no role in your study, please state: “The funders had no role in study design, data collection and analysis, decision to publish, or preparation of the manuscript.”

2. Figure [3]: please (a) provide a direct link to the base layer of the map used and ensure this is also included in the figure legend; (b) provide a link to the terms of use / license information for the base layer. We cannot publish proprietary or copyrighted maps (e.g. Google Maps, Mapquest) and the terms of use for your map base layer must be compatible with our CC-BY 4.0 license.

3. In the online submission form, you indicated that "Data can be made available on request. Suppl File 2 provides much of the details.". All PLOS journals now require all data underlying the findings described in their manuscript to be freely available to other researchers, either 1. In a public repository, 2. Within the manuscript itself, or 3. Uploaded as supplementary information.

4. Please provide separate figure files in .tif or .eps format and removed from the manuscript file.

Additional Editor Comments (if provided):

Reviewers' comments:

Reviewer's Responses to Questions

**Comments to the Author**

1. Does this manuscript meet PLOS Global Public Health’s publication criteria? Is the manuscript technically sound, and do the data support the conclusions? The manuscript must describe methodologically and ethically rigorous research with conclusions that are appropriately drawn based on the data presented.

Reviewer #1: Yes

Reviewer #2: Yes

2. Has the statistical analysis been performed appropriately and rigorously?

Reviewer #1: N/A

Reviewer #2: N/A

3. Have the authors made all data underlying the findings in their manuscript fully available (please refer to the Data Availability Statement at the start of the manuscript PDF file)?

Reviewer #1: No

Reviewer #2: No

4. Is the manuscript presented in an intelligible fashion and written in standard English?

Reviewer #1: Yes

Reviewer #2: Yes

5. Review Comments to the Author

Reviewer #1: This review has been carried out well. The authors have presented all key aspects of the review process and the results.

There are a few comments for revision which are as mentioned below.

1.Line 133: Mention the last date of search based on which the review was carried out.

2.Line 179-187: The calculation of numbers of references are wrong. Example: Out of total 6595 references, removing 821 and 634 references will result is 5140 references. But, it mentioned as 5136. Similarly, the final number of items comes to 179 as per the description but it is mentioned as 175. These need to be corrected.

Reviewer #2: The topic is interesting. The role of the midwives in ensuring continuity of care and in preventing pre, per and postnatal complications should be highlighted and well recognized and adopted in the healthcare system.

I would recommend reflecting on the factors that impede the continuity of care provided by the midwives in the introduction.

Since eligible papers include care providers who are midwives and non-midwives, such as, nurses, community health workers and physicians, I would suggest changing the title to pre, per, postnatal care instead of midwifery care. Continuity of care may be provided by cadre other than midwives as per the provided definition, but this might be confusing if you want to focus on the role of the midwife per se. But it is fine if the focus is on the continuity of maternal and newborn care.

Kindly justify why the literature search was limited to the last 10 years only.

Kindly specify the percentage of the health providers per specialty. This will give an idea about who are the providers of care.

I wonder if you can provide more details about the profile/qualifications of the healthcare providers, the way the midwifery system function, if any affiliation to healthcare centres, support system, Health cost coverage, Safety, Outcome indicators, etc.

Make sure the paper is edited.

6. PLOS authors have the option to publish the peer review history of their article (what does this mean?). If published, this will include your full peer review and any attached files.

**Do you want your identity to be public for this peer review?** For information about this choice, including consent withdrawal, please see our Privacy Policy.

Reviewer #1: **Yes: **DEEPANJALI BEHERA

Reviewer #2: **Yes: **Mathilde Azar

---

## [Decision Letter · Decision Letter 1]

7 Sep 2022

Midwifery continuity of care: A scoping review of where, how, by whom and for whom?

PGPH-D-22-00905R1

Dear Professor Homer,

We are pleased to inform you that your manuscript 'Midwifery continuity of care: A scoping review of where, how, by whom and for whom?' has been provisionally accepted for publication in PLOS Global Public Health.

Best regards,

Ahmed Waqas

Academic Editor

Reviewer Comments (if any, and for reference):

Reviewer's Responses to Questions

**Comments to the Author**

1. If the authors have adequately addressed your comments raised in a previous round of review and you feel that this manuscript is now acceptable for publication, you may indicate that here to bypass the “Comments to the Author” section, enter your conflict of interest statement in the “Confidential to Editor” section, and submit your "Accept" recommendation.

Reviewer #1: All comments have been addressed

Reviewer #2: (No Response)

2. Does this manuscript meet PLOS Global Public Health’s publication criteria? Is the manuscript technically sound, and do the data support the conclusions? The manuscript must describe methodologically and ethically rigorous research with conclusions that are appropriately drawn based on the data presented.

Reviewer #1: Yes

Reviewer #2: (No Response)

3. Has the statistical analysis been performed appropriately and rigorously?

Reviewer #1: N/A

Reviewer #2: (No Response)

4. Have the authors made all data underlying the findings in their manuscript fully available (please refer to the Data Availability Statement at the start of the manuscript PDF file)?

Reviewer #1: No

Reviewer #2: Yes

5. Is the manuscript presented in an intelligible fashion and written in standard English?

Reviewer #1: Yes

Reviewer #2: Yes

6. Review Comments to the Author

Reviewer #1: The article appears sound after revision.

Reviewer #2: The authors addressed some of the comments. The others might not have been documented to answer. But this does not affect the quality of the paper or the aim of the study. So, we can ignore them.

7. PLOS authors have the option to publish the peer review history of their article (what does this mean?). If published, this will include your full peer review and any attached files.

**Do you want your identity to be public for this peer review?** For information about this choice, including consent withdrawal, please see our Privacy Policy.

Reviewer #1: **Yes: **Deepanjali Behera

Reviewer #2: **Yes: **Mathilde Azar
